# The Effects of Dietary Education Interventions on Individuals with Type 2 Diabetes: A Systematic Review and Meta-Analysis

**DOI:** 10.3390/ijerph18168439

**Published:** 2021-08-10

**Authors:** Juri Kim, Myung-Haeng Hur

**Affiliations:** 1Department of Nursing, University of Kyungmin, 545 Seobu-ro, Uijeongbu-si 11618, Korea; kimjui200@gmail.com; 2College of Nursing, Eulji University, 712 Dongil-ro, Uijeongbu-si 11759, Korea

**Keywords:** diabetes mellitus, diet, education, systematic review

## Abstract

As the incidence and prevalence of diabetes increases, intervention through dietary education is becoming more important for diabetes control. This systematic review examines the evidence for the efficacy of dietary education interventions on diabetes control. The study subjects were patients with type 2 diabetes, and the main outcome variable was glycosylated hemoglobin level (HbA1c). The target studies were randomized controlled trials. Thirty-six studies were included in the analysis, of which 33 were included in the meta-analysis. The effect size between dietary education and general interventions was −0.42 (*n* = 5639, MD = −0.42; 95% CI −0.53 to −0.31) and was significantly different (Z = 7.73, *p* < 0.001). When subgroup analyses were performed following the application periods, intervention methods, and intervention contents, the mean differences in 4–6-month application, individual education, and diet-exercise-psychosocial intervention were −0.51, (*n* = 2742, 95% CI −0.71 to −0.32), −0.63 (*n* = 627, 95% CI −1.00 to −0.26), and −0.51 (*n* = 3244, 95% CI −0.71 to −0.32), respectively. Dietary education interventions provided for at least 3 months were highly effective in controlling HbA1c levels. Regarding the education method, individualized education was more effective, and contact or non-contact education may be applied for this. Combining diet, exercise, and psychosocial intervention is more effective than diet education alone.

## 1. Introduction

The incidence of diabetes mellitus is increasing worldwide. According to the International Diabetes Association, diabetes patients worldwide account for 8.3% of the total population, and it is expected that this number will reach 592 million by 2035 [1]. Diabetes is a chronic metabolic disease that causes complications such as cardiovascular disease, arteriosclerosis, hypertension, neuropathy, nephropathy, and diabetic retinopathy [2]. Type 2 diabetes usually occurs after the age of 40 and accounts for about 90% of all diabetes patients. Unlike type 1 diabetes, there are often no clear clinical symptoms in the early stages. The onset of type 2 diabetes is preceded by a decrease in insulin secretion, followed by a metabolic disorder due to an increase in insulin resistance [3]. In many cases, type 2 diabetes mellitus can be improved if weight is reduced through diet and exercise at an early stage [3].

Thus, the most basic goal of treatment is to maintain a normal blood glucose level [2]. Treatment options include insulin therapy, exercise, dietary intervention, and psychological intervention [2]. Although drug and insulin therapy are necessary, patient-centered dietary and exercises education interventions to prevent complications have also gained importance. Dietary education is essential and requires education, counseling, and diet management [4]. Previous studies have reported that dietary education interventions can cause a significant reduction in not only body mass index (BMI), glycated hemoglobin (HbA1c), and fasting blood sugar levels, but also the risk of microvascular complications and cardiovascular disease [5,6].

To the authors’ knowledge, a total of five systematic reviews of the effectiveness of dietary education interventions for patients with type 2 diabetes have been published previously. In these reviews, the intervention methods included remote therapy intervention, web education intervention, dietary carbohydrate restriction, and nutrition therapy [7,8,9,10,11]. However, dietary education is provided in various forms such as individualized, group, self-help group, and web-based education, and different content focusing on general or specific diet information is provided. Moreover, the duration of education varies with each study; thus, it is necessary to assess the effects of each of these aspects. Outcome variables to assess the effects of dietary education on blood glucose control were body weight, BMI, blood pressure, postprandial blood glucose, glycosylated hemoglobin level (HbA1c), and cholesterol; the most important outcome variable was HbA1c. HbA1c, an important indicator of glycemic control, is closely correlated with average blood sugar levels in diabetes patients, and it is also an indicator for the reduced risk of complications [12,13]. Thus, HbA1_C_ is a suitable indicator to assess the clinical effects of dietary education.

This study aimed to systematically review randomized controlled trials (RCTs) that assessed HbA1c levels after providing dietary education interventions in various ways and with different content. Then, a meta-analysis was performed to estimate the effects of dietary education interventions in patients with type 2 diabetes.

## 2. Materials and Methods

### 2.1. Data Sources and Searches 

In this study, we systematically reviewed RCTs that provided qualitative and quantitative data to assess the clinical effects of dietary education interventions in patients with type 2 diabetes. Literature searches were conducted up to March 2020 using international databases such as PubMed, EMBASE, CINAHL, and Cochrane Central Register of Controlled Trials. The domestic databases used to search for published journals and theses were DBpia, Korean Studies Information Service System, Research Information Service System, NDSL, and Korea Med. To increase the sensitivity of the literature search, gray literature such as theses, news, and presentations were searched for manually in addition to searching in electronic databases. MeSH terms and text words using AND/OR and truncation were used for the literature search. MeSH terms were used to search for articles in international databases. Studies that used the term “diabetes mellitus” as the intervention method were selected, and main variables such as “diet” and “education” terms were used for extraction. Filtering was used according to the characteristics of each database, and methods to increase the specificity and sensitivity of the search were used. The search terms used in the international databases were “Diabetes mellitus” and “Education” and “Diet” and “HbA1c.” The MeSH search function was not available in the domestic databases. Thus, concept words for “type 2 diabetes,” “metabolic syndrome,” “blood sugar control,” and “diet” were used as keywords to search for articles (Table 1). Research reports and theses were excluded from the study. This study was exempt from needing approval from the institutional review board as it is a systematic review (EUIRB2020-08).

### 2.2. Study Selection

Two investigators (the first and second authors) independently evaluated the articles for eligibility. Studies on the effects of dietary education interventions in type 2 diabetes patients, especially RCTs, were selected. Academic papers were chosen when there was an overlap between academic papers and a thesis. Participants, intervention, comparison, and outcome (PICO), which are the specific questions for systematic literature reviews, are as follows. The patient population of this study included adults with type 2 diabetes. Experimental interventions considered for this study were web-based, individualized or grouped, or self-help group dietary education interventions or diet-related educational interventions. The comparative intervention in this study was the general education intervention provided to diabetes patients. In this study, the control group was the group that received usual care provided to diabetic subjects. The outcome after the intervention in type 2 diabetes patients was blood glucose level. In this study, HbA1c, which is representative of the blood glucose level for the last 3 months, was assessed.

### 2.3. Inclusion and Exclusion Criteria 

This systematic review considered for inclusion any RCT that assesses a Dietary Education intervention in type 2 diabetes mellitus population. The primary outcomes were HbA1c at different follow-up periods to measure the glycemic control of type 2 diabetes mellitus. Exclusion criteria were: (1) study designs other than RCTs; (2) type 1 diabetes mellitus population, prediabetes, gestational diabetes mellitus; (3) educational interventions that do not include dietary interventions; (4) Studies not published in Korean or English; (5) unpublished theses; and (6) experiments on animals or studies on children, preclinical studies.

### 2.4. Data Synthesis and Analysis

Cochrane Review Manager (RevMan, London, UK) software 5.3 was used for the analysis of selected studies that measured outcome variables and systematic intervention methods. Meta-analysis can be performed when multiple scientific studies address the same question, with each study reporting measurements that are expected to have some degree of error. The Cochrane Quality Assessment tool in Cochrane software evaluates the risk of selection bias, performance bias, detection bias, attrition bias, and reporting bias as low, high, or uncertain. Results were then entered into RevMan and evaluation results for the assessed risk were presented according to the evaluation criteria. 

## 3. Results

### 3.1. Characteristics of Studies Selected for the Systematic Literature Review

Published journals were searched in domestic and international databases until March 2020, and gray journals were searched manually. A total of 36 studies were selected and included in the analysis (Table 1). The meta-analysis was performed on the 33 selected papers, wherein the outcome variables and intervention methods were the same. Of the 33 pieces of literature available for meta-analysis, 5 studies provided only dietary education interventions, the other 9 studies provided dietary education interventions and exercise therapy, and the remaining 19 studies provided dietary education interventions, exercise therapy, and psychosocial therapy. The results and number of experimental and control groups are presented in Table 1 and Figure 1.

### 3.2. Literature Quality Assessment 

As a result of the quality evaluation of the study, there were several cases with unclear performance and detection bias; the attrition bias was due to the high attrition rate of the study participants. The results are shown in Figure 2.

### 3.3. Effects of Dietary Education Interventions on HbA1cin Type 2 Diabetes Patients

Among the selected 36 papers, the effect size on the dietary education of type 2 diabetes patients was meta-analyzed for 33 of which the effect size analysis was possible. In addition, subgroup analysis was performed according to education period, education method, and education content.

#### 3.3.1. Comparison of the HbA1_C_ Effect Size According to the Duration of Dietary Education Interventions in Type 2 Diabetes Patients 

The effect size of HbA1_C_ in type 2 diabetes patients was analyzed by dividing it into the type of educational intervention and follow-up time points.

##### HbA1_C_ Effect Size at the Endpoints of Dietary Education Interventions 

The effect size in the dietary education experimental group decreased by 0.28 (*n* = 385, MD = −0.28; 95% CI −0.65 to 0.09) compared with the control group; however, the difference was not significant (Z = 1.51, *p* < 0.13) (Figure 3).

An analysis of 14 studies that assessed HbA1c 4–6 months after the completion of dietary education showed that the effect size in the dietary education experimental group decreased by 0.51 (*n* = 2742, MD = −0.51; 95% CI −0.71 to −0.32) compared with the control group. The difference in the effect size between the two groups was significant (Z = 5.18, *p* < 0.001).

An analysis of three studies that assessed HbA1c 7–9 months after dietary education completion showed that the effect size in the dietary education experimental group decreased by 0.40 (*n* = 511, MD = −0.40; 95% CI −0.43 to −0.37) compared with that seen in the control group. The difference in the effect size between the two groups was significant (Z = 26.14, *p* < 0.001).

An analysis of 11 studies that assessed HbA1c 10–12 months after the completion of dietary education showed that the effect size in the dietary education experimental group decreased by 0.41 (*n* = 1998, MD = −0.41; 95% CI −0.62 to −0.21) compared with that in the control group. The difference in the effect size between the two groups was significant (Z = 3.93, *p* < 0.001).

##### HbA1c Effect Size in Dietary Education Interventions Assessed at Different Follow-up Time-Points

After completion of the dietary education intervention in type 2 diabetes patients, the effects were analyzed at different HbA1c measurement durations (Figure 4). An analysis of nine studies showed that the effect size of HbA1c during dietary education intervention for 3 months was not homogeneous (Education G. vs. Control group G.: I_2_ = 83%). Therefore, a random-effects model was used to analyze the results between the experimental and control groups; the effect size decreased by 0.32 (*n* = 672, MD = −0.32; 95% CI −0.59 to −0.05) in the experimental group compared with the control group, which was not significant (Z = 2.31, *p* = 0.02).

An analysis of 25 studies that assessed HbA1_C_ during dietary education for 4–6 months showed that the effect size of HbA1_C_ in the experimental group decreased by 0.47 (*n* = 3915, MD = −0.47, 95% CI −0.63 to −0.30) compared with that in the comparison group; the difference was significant (Z = 5.58, *p* < 0.001).

An analysis of three studies that assessed HbA1_C_ during dietary education for 7–9 months showed that the effect size of HbA1_C_ in the experimental group decreased by 0.40 (*n* = 511, MD = −0.40; 95% CI −0.43 to −0.37) compared with the control group; the difference was significant (Z = 26.16, *p* < 0.001).

An analysis of HbA1_C_ during dietary education for 10–12 months showed that the effect size of HbA1_C_ in the experimental group decreased by 0.46 (*n* = 2600, MD = −0.46; 95% CI −0.66 to −0.27) compared with the control group; the difference was significant (Z = 4.60, *p* < 0.001).

#### 3.3.2. Comparison of the Effect Size of HbA1_C_ According to Dietary Education Intervention Methods

The HbA1_C_ effect size was analyzed according to different methods of dietary education in patients with type 2 diabetes (Figure 5). The intervention methods were divided into non-face-to-face and face-to-face education. Ten studies used web- and mobile-phone-based non-face-to-face education and face-to-face education was classified into individualized and grouped education interventions. There were four and 19 studies on individualized and grouped education interventions, respectively.

An analysis of 10 studies on the web- and mobile phone-based non-face-to-face dietary education interventions showed that the effect size of HbA1_C_ in the experimental group decreased by 0.42 (*n* = 2282, MD = −0.42 95% CI −0.65 to −0.18) compared with the control group, and the difference was significant (Z = 3.45, *p* = 0.006).

The effect size in the experimental group that received individual dietary education interventions decreased by 0.63 (*n* = 627, MD = −0.63; 95% CI: −1.00.to −0.26) compared with the control group, and the difference was significant (Z = 3.33, *p* < 0.009). The effect size in the experimental group that received grouped dietary education interventions decreased by 0.38 (*n* = 2727, MD = −0.38; 95% CI −0.52 to −0.24) compared with the control group, and the difference was significant (Z = 5.23, *p* < 0.001). 

#### 3.3.3. Comparison of the Effect Size of HbA1_C_ According to Dietary Education Contents

There were five studies on dietary-centered education interventions. Interventions in the group included a low-carbonate group, low-fat group, low glycemic index (GI) diet group, and low-fruit group vs. high-fruit group. There were nine dietary and exercise education interventions and 19 studies on dietary, exercise, and psychosocial education interventions (Figure 6).

A comparison between the experimental and control groups that underwent dietary-centered education interventions and general interventions, respectively, showed that the effect size decreased by 0.15 (*n* = 568, MD = −0.15; 95% CI −0.46 to 0.17) in the experimental group compared with the control group, which was not significant (Z = 0.92, *p* = 0.36).

Comparison between the experimental and control groups that underwent dietary and exercise education intervention and general intervention, respectively, showed that the effect size decreased by 0.48 (*n* = 1808, MD = −0.48; 95% CI −0.73 to −0.24) in the experimental group compared with the comparison group, which was significant (Z = 3.85, *p* < 0.001). 

A comparison between the experimental and control groups that underwent dietary exercise and a psychosocial education intervention and a general intervention, respectively, showed that the effect size decreased by 0.48 (*n* = 3260, MD = −0.48; 95% CI −0.61 to −0.35 in the experimental group compared with the comparison group, and the difference was significant (Z = 7.21, *p* < 0.001).

### 3.4. Publication Bias

The 33 studies analyzed in the meta-analysis are scattered around the effect estimate, whereas the large-scale studies are distributed at the top of the graph. Small-scale studies are distributed at the bottom of the graph and the graph is shaped like a funnel; it indicates that there is no publication bias (Figure 7).

## 4. Discussion

This systematic literature review was performed to assess the effects of dietary education interventions in type 2 diabetes patients. The interventions included web-based, self-help, and individualized or grouped dietary or educational interventions that included diet. Comparison groups consisted of type 2 diabetes patients who were provided with general education interventions. HbA1c was selected as the outcome variable. 

We found that HbA1c levels were lower in the experimental group after dietary education interventions compared with those in the control group. Diet and exercise interventions are emphasized as important in diabetes guidelines [35] and nutritional interventions are effective in controlling blood sugar levels [11]. Therefore, dietary education interventions for diabetes patients are thought to be effective interventions for controlling blood sugar levels. Subgroup analysis was performed to analyze HbA1c levels according to the duration of dietary education. HbA1c levels assessed after 4–6, 7–9, and 10–12 months of dietary education interventions were lower in the experimental group than in the control group. In contrast, HbA1c assessed after 3 months of an education intervention did not show a significant difference between the two groups. In addition, HbA1_C_ levels according to the duration of dietary education intervention, including the follow-up period, were lower in the experimental group at 4–6, 7–9, and 10–12 months. Studies have shown that a repetitive and long-term dietary education intervention that offers follow-up management was more effective than a short-term education intervention [35]. Moreover, considering that HbA1c reflects the blood sugar level at 3 months, it is thought that dietary education interventions for 4 months or longer are necessary. In particular, there is a need for continuous control of blood glucose levels in diabetes to prevent complications. Maintaining HbA1c levels < 6.5% for 6 years is known to help prevent complications, including microvascular complications [50]. Therefore, continuous follow-up interventions would be necessary in addition to dietary education interventions for 4 months or longer.

Analysis of face-to-face and web- and mobile phone-based non-face-to-face education interventions showed that both face-to-face (individualized and group education) and non-face-to-face interventions were effective. In particular, individual education interventions showed low heterogeneity between studies and large effect sizes. Experimental studies reported that HbA1c decreased by 1.0%–2.0% in type 1 and type 2 diabetes patients after individual nutritional education [51,52]. Furthermore, systematic reviews have shown that web-based education interventions led to decreased HbA1_C_ [8,9]. Considering these findings, individual education seems to be effective and should be given to diabetes patients.

Different dietary education intervention contents were also analyzed. Subgroup analysis of a dietary-centered education intervention, dietary and athletic education intervention, and dietary exercise and psychosocial intervention showed that the effect size of HbA1c was significantly reduced in the two intervention groups, except for the dietary-centered education intervention. This finding is consistent with the results of a 20-year follow-up study, which showed significantly decreased HbA1_C_ after a dietary and athletic education interventions [53]. Similar findings were reported by another study where HbA1c significantly decreased after a dietary education and moderate exercise strategy that included a decrease of 500–750 kcal and 175 min of moderate-intensity exercise per week. Therefore, combining diet, exercise, and psychosocial intervention is considered more effective than diet education alone.

The limitations of this study are as follows. First, the contents and methods of interventions were diverse between studies, and it was difficult to divide them into different groups. Second, the literature search was limited to articles published in academic journals. Thus, research reports and theses were excluded. Moreover, a tendency to publish only statistically significant results was noted. Therefore, there may be a possibility of publication bias or overestimation of the results. Another limitation is that the results of the sub-analysis on dietary education interventions could not be derived as intervention methods and detailed contents were not provided in studies. The studies included in this meta-analysis have included complex dietary education interventions, so there may be high heterogeneity. Therefore, it is necessary to carefully interpret the research results. 

## 5. Conclusions

Dietary education interventions are very effective in controlling blood sugar, and a duration of at least 3 months is required. Individual education is more effective than face-to-face or web and mobile phone education. Further, interventions are thought to be more effective when dietary, exercise, and psychosocial education interventions are provided together rather than when dietary education is provided alone. Studies with long-term as opposed to short-term interventions are needed; web- and mobile-based individual dietary education interventions would be more effective than group interventions. Further research is necessary to present a wide range of generalized results, including the specific variables in the study.

## Figures and Tables

**Figure 1 ijerph-18-08439-f001:**
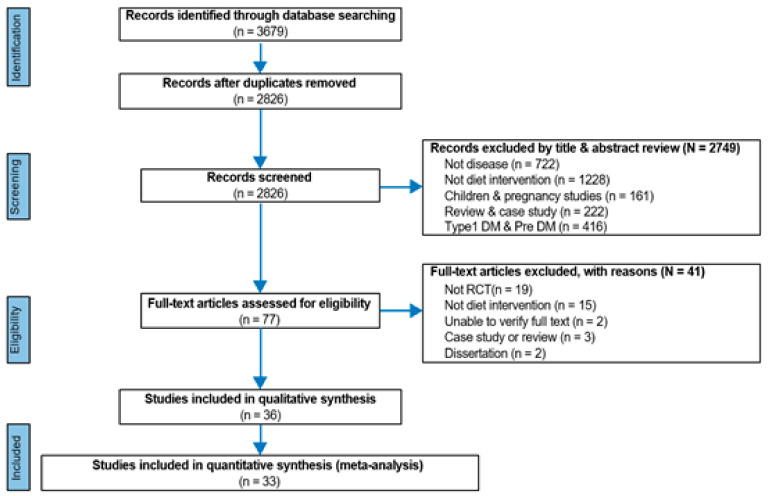
Flowchart of the study selection process. DM = Diabetes mellitus, RCT = Randomized controlled trial.

**Figure 2 ijerph-18-08439-f002:**
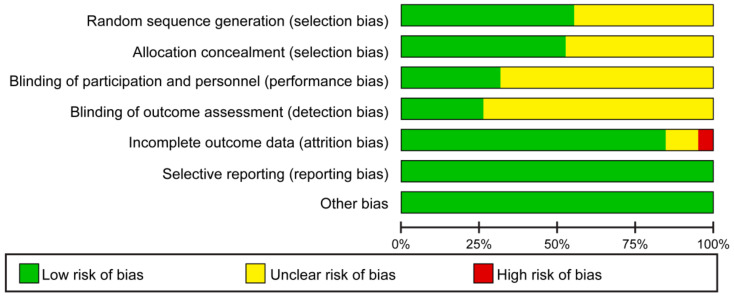
Risk of bias graph.

**Figure 3 ijerph-18-08439-f003:**
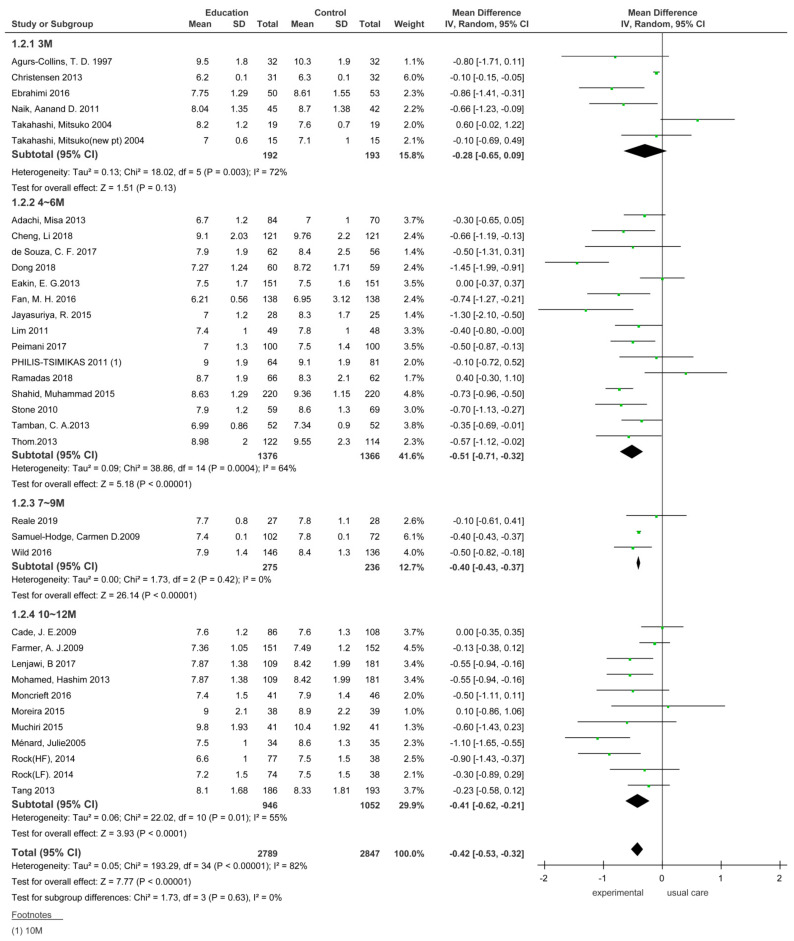
Immediate effect of dietary education intervention on patients with type 2 diabetes. SD, standard deviation; CI, confidence interval; IV, inverse variance.

**Figure 4 ijerph-18-08439-f004:**
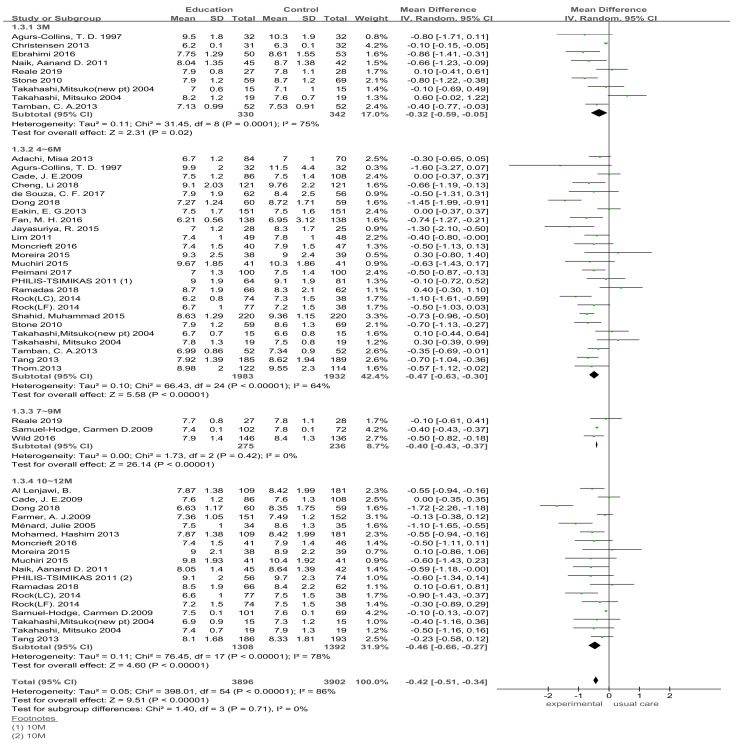
Follow-up effect of dietary education intervention on patients with type 2 diabetes. SD, standard deviation; CI, confidence interval; IV, inverse variance.

**Figure 5 ijerph-18-08439-f005:**
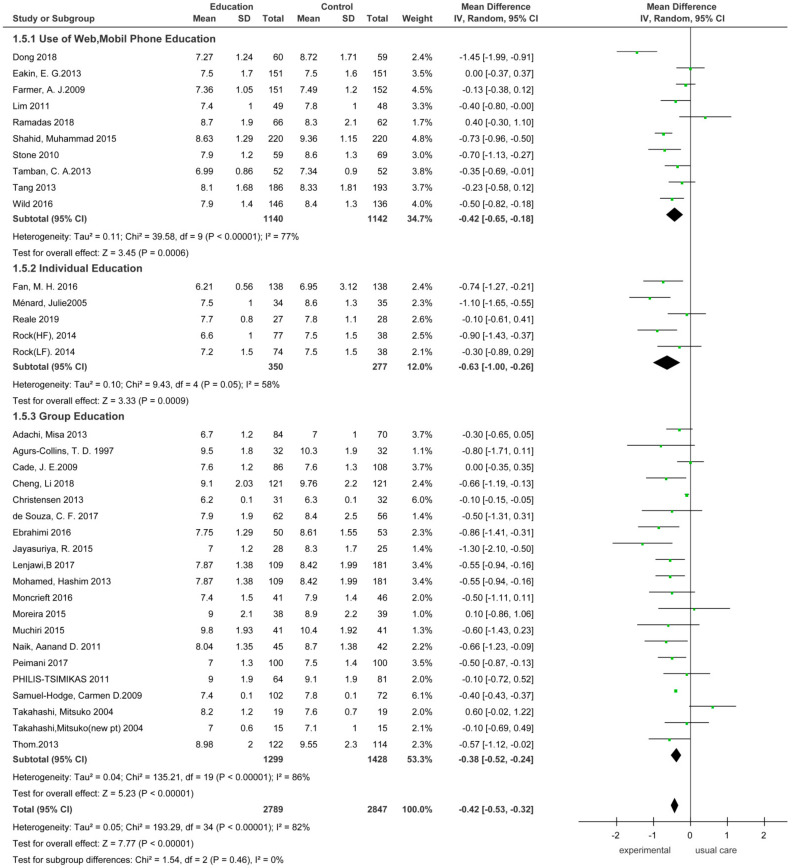
Forest plot of the effect of the intervention method. SD, standard deviation; CI, confidence interval; IV, inverse variance.

**Figure 6 ijerph-18-08439-f006:**
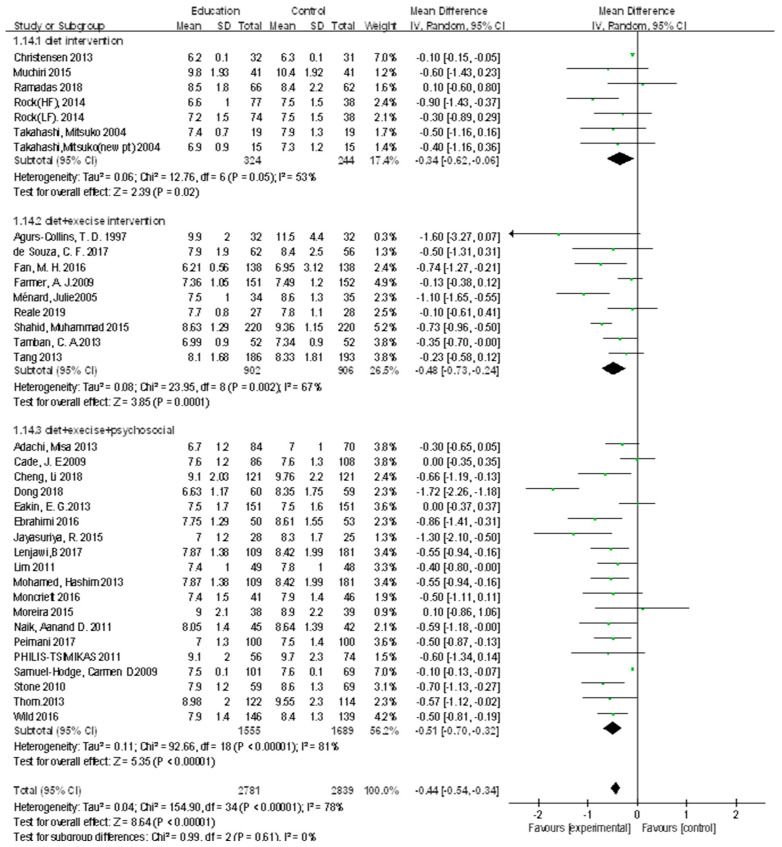
Forest plot of the effect of diet education. SD, standard deviation; CI, confidence interval; IV, inverse variance.

**Figure 7 ijerph-18-08439-f007:**
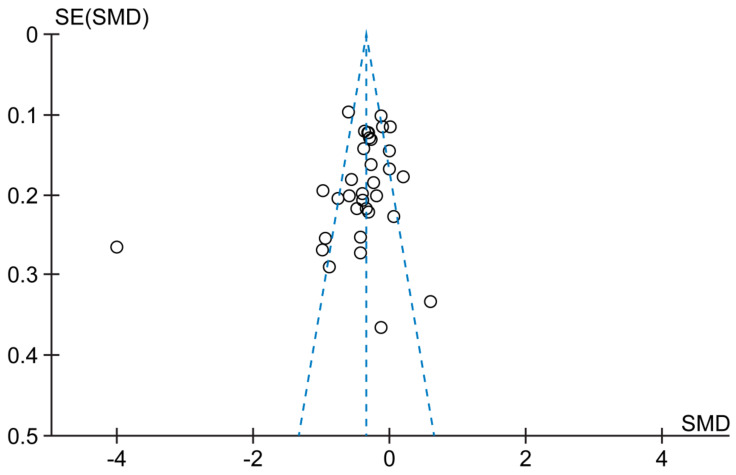
Funnel plot of comparison. SE: Standard Error, SD: standard deviation.

**Table 1 ijerph-18-08439-t001:** Summary of randomized controlled trials on the effects of diet education for patients with type 2 diabetes mellitus.

First Author (yr)/ Country	Intervention Group Intervention Method	Control Group	Age (yr) M ± SD or Median (IQR)	Length of Program	Result	Authors’ Conclusions
* Adachi (2013) Japan [14]	(A) Structured individual-based lifestyle education (SILE) program (*n* = 84)	(B) Control group (*n* = 70)	(A) 60.4 ± 11.4 (B) 62.3 ± 10.1	6 M	MD −0.30 95% CI [−0.65, 0.05]	The SILE program that was provided in primary care settings for patients with type 2 diabetes resulted in greater improvement in HbA1c levels than usual diabetes care and education.
* Agurscollins (1997) USA [15]	(A) Intervention (*n* = 32)	(B) Control group (*n* = 32)	(A) 62.4 ± 5.9 (B) 61.0 ± 5.7	3 M/6 M	MD −0.80 95% CI [−1.71, 0.11]	The decrease in HbA1c values was generally independent of the relatively modest changes in dietary intake, weight, and activity and may reflect indirect program effects on other aspects of self-care.
* Cade (2009) Canada [16]	(A) Peer Expert Patient Program (EPP) (*n* = 86)	(B) Control group (*n* = 108)	(A) 65.4 ± 11.6 (B) 66.2 ± 11.5	6 M/12 M	MD 0.00 95% CI [−0.35, 0.35]	The EPP approach was not effective in changing measures of diabetes control or diet.
* Cheng (2018) China [17]	(A) Empowerment-based self-management program (*n* = 121)	(B) Control group (*n* = 121)	(A) 56.13 ± 10.72 (B) 53.91 ± 13.01	5 M	MD −0.66 95% CI [−1.19, −0.13]	Findings indicate that the patient-centered, empowerment-based self-management intervention program did not induce a significant HbA1c reduction.
* Christensen (2013) Denmark [18]	(A) Low-fruit (*n* = 31)	(B) High-fruit (*n* = 32)	(A) 57 ± 12 (B) 59 ± 12	3 M	MD −0.10 95% CI [−0.15, −0.05]	HbA1c decreased in both groups with no difference between the groups (difference: 0.19%, 95% CI: −0.23 to 0.62).
* Dong (2018) China [19]	(A) Health education using the WeChat platform plus usual care (*n* = 60)	(B) Control group (*n* = 59)	NR	6 M/12 M	MD −1.72 95% CI [−1.99, −0.91]	Health education of diabetic individuals via the WeChat platform in conjunction with conventional diabetes treatment could improve glycemic control and positively influence other aspects of diabetes self-care skills.
* Eakin (2013) USA [20]	(A) Telephone counseling (*n* = 151)	(B) Control group (*n* = 151)	(A) 57.7 ± 8.1 (B) 58.3 ± 9.0	6 M	MD 0.00 95% CI [−0.37, 0.37]	No intervention effect for HbA1c (RR = 0.99, 95% CI: 0.96, 1.01).
* Ebrahimi (2016) Iran [21]	(A) Empowerment model (*n* = 50)	(B) Control group (*n* = 53)	(A) 46.97 ± 5.54 (B) 48.15 ± 6.52	3 M	MD −0.86 95% CI [−1.41, −0.31]	Study results indicated the positive effects of applying the empowerment model on the metabolic control indicators.
Etienne (2017) Rwanda [22]	(A) Lifestyle education program (*n* = 115)	(B) Control group (*n* = 108)	(A) 51.4 ± 10.9 (B) 50.5 ± 11.0	12 M	NR	This study demonstrated that a structured lifestyle group education program for people with diabetes is an attractive option in a resource-limited setting, as it showed significant benefits in improved glycemic control over 12 months.
* Fan (2016) Australia [23]	(A) Individualized education (*n* = 138)	(B) Control group (*n* = 138)	(A)62.94 ± 10.72 (B) 64.89 ± 10.14	6 M	MD −0.74 95% CI [−1.27, −0.21]	Individualized diabetes education is more effective than group education in facilitating the control of type 2 diabetes.
* Farmer (2009) UK [24]	(A) Intensive self-monitoring blood glucose (*n* = 151)	(B) Control group (*n* = 152)	(A) 65.5 ± 9.9 (B) 66.3 ± 10.2	12 M	MD −0.13 95% CI [−0.38, 0.12]	Significant improvement in glycemic control compared with usual care monitored by HbA1c levels.
* Jayasuriya (2015) Australia [25]	(A) Diabetes Self-Management (DSM) Intervention (*n* = 28)	(B) Control group (*n* = 25)	(A) 51.5 ± 7.5 (B) 51.4 ± 7.1	6 M	MD −1.30 95% CI [−2.10, −0.50]	There was a significant difference in HbA1c between the groups.
* Lim (2011) South of Korea [26]	(A) Based ubiquitous healthcare service (*n* = 49)	(B) Control Group (*n* = 48)	(A) 67.2 ± 4.1 (B) 68.1 ± 5.5	6 M	MD −0.40 95% CI [−0.80, −0.00]	U-healthcare service achieved better glycemic control with less hypoglycemia than SMBG (self-monitored blood glucose) and routine care and may provide effective and safe diabetes management in elderly diabetic patients.
* Lenjawi (2017) Qatar [27]	(A) Nurse-led, group-based diabetes educational program (*n* = 109)	(B) Control group (*n* = 181)	(A) 52 ± 8.9 (B) 55 ± 9.7	12 M	MD −0.55 95% CI [−0.94, −0.16]	The inclusion of South Asian patients with type II diabetes in a structured, theory-based diabetes educational program that is led by nurses improves glycemic and metabolic parameters after 12 months.
* Ménard (2005) Canada [28]	(A) Intensive multi therapy (*n* = 34, 32)	(B) Control group (*n* = 35, 29)	(A) 53.7 ± 7.5 (B) 55.9 ± 8.6	12 M/18 M	MD −1.10 95% CI [−1.65, −0.55]	Successful in helping patients meet most of the goals set by a national diabetes association. However, 6 months after intensive therapy stopped and patients returned to the control group, the benefits had vanished.
* Mohamed (2013) Qatar [29]	(A) Culturally sensitive, structured education program (CSSEP) (*n* = 109)	(B) Control group (*n* = 181)	(A) 52 ± 8.9 (B) 55 ± 10.7	12 M	MD −0.55 95% CI [−0.94, −0.16]	After 12 months of participation, the intervention was shown to have led to a statistically significant reduction in HbA1_C_ in the CSSEP group.
* Muchiri (2015) South Africa [30]	(A) Nutrition education sessions (*n* = 41)	(B) Control group (*n* = 41)	(A) 59.4 ± 6.9 (B) 58.2 ± 8.0	6 M/12 M	MD −0.6 95% CI [−1.43, 0.23]	Nutrition education was not efficacious on HbA1c.
* Moreira (2015) Brazil [31]	(A) Nursing case management (*n* = 38)	(B) Control group (*n* = 39)	(A) 50.0 ± 6.5 (B) 50.3 ± 7.6	6 M/12 M	MD 0.10 95% CI [−0.86, 1.06]	Both groups showed a statistically significant reduction in HbA1c at 6- and 12-months following baseline.
* Moncrieft (2016) USA [32]	(A) Lifestyle intervention (*n* = 55, 40, 41)	(B) Control group (*n* = 51, 47, 46)	(A) 54.8 ± 8.27 (B) 54.8 ± 6.34	6 M/12 M	MD −0.50 95% CI [−1.11, 0.11]	Multicomponent behavioral interventions targeting weight loss and depressive symptoms as well as diet and physical activity are efficacious in the management of Type 2 diabetes.
* Naik (2011) USA [33]	(A) Empowering Patients in Care (EPIC) (*n* = 45)	(B) Control group (*n* = 42)	(A) 63.82 ± 7.9 (B) 63.45 ± 7.8	3 M/12 M	MD −0.66 95% CI [−1.23, −0.09]	Primary care-based DM group clinics that include structured goal-setting approaches to self-management can significantly improve HbA1c levels after intervention and maintain improvements for 1 year.
* Peimani (2017) Iran [34]	(A) Peer support intervention (*n* = 100)	(B) Control group (*n* = 100)	(A) 59.0 ± 11.3 (B) 58.8 ± 11.7	6 M	MD −0.50 95% CI [−0.87, −0.13]	Peer support activities can be successfully applied in diabetes self-management, especially in areas with a shortage of professionals and economic resources.
* Philistsimkas (2011) USA [35]	(A) Trained peer education (*n* = 104)	(B) Control group (*n* = 103)	(A) 52.2 ± 9.6 (B) 49.2 ± 11.8	4 M/10 M	MD 0.00 95% CI [−0.62, 0.62]	The Project Dulce model of culturally sensitive, peer-led education, demonstrates improvement in glucose and metabolic control and suggests that this low-cost approach to self-management education for high-risk diabetic populations is effective.
* Ramadas (2018) Malaysia [36]	(A) Web-based dietary (*n* = 66)	(B) Control group (*n* = 62)	(A) 49.6 ± 10.7 (B) 51.5 ± 10.3	6 M/12 M	MD 0.40 95% CI [−0.30,1.10]	Aided by improvements in knowledge and attitudes.
* Rock (2014) USA [37]	(A) Low fat (*n* = 74) (B) High fat (*n* = 77)	(C) Control group (*n* = 76)	(A) 55.5 ± 9.2 (B) 57.3 ± 8.6 (C) 56.8 ± 9.3	6 M/12 M	LF MD −0.30 95% CI [−0.89, 0.29] HF MD −0.90 95% CI [−1.43, −0.37]	The weight loss program resulted in greater weight loss and improved glycemic control in type 2 diabetes patients.
* Reale (2019) Italy [38]	(A) Individual education (IE) (*n* = 27)	(B) Control group (*n* = 28)	(A) 59.4 ± 9.1 (B) 61.5 ± 8.2	3 M/8 M	MD −0.10 95% CI [−0.61, 0.41]	Our trial provides preliminary data regarding the efficacy of structured group and individual education on achieving better neurometabolic control without drug therapy reinforcement and with positive effects on patients’ attitudes and treatment satisfaction.
* Samuelhodge (2009) USA [39]	(A) Church-based diabetes self- management (*n* = 102/101)	(B) Control group (*n* = 72/69)	(A) 57.0 ± 0.9 (B) 61.3 ± 1.3	8 M/12 M	MD −0.40 95% CI [−0.43, −0.37]	At 12 months, the difference between groups was not significant. The church-based intervention was well received by participants and improved short-term metabolic control.
* Shahid (2015) Pakistan [40]	(A) Mobile phone intervention (*n* = 220)	(B) Control group (*n* = 220)	(A) 48.95 ± 8.83 (B) 49.21 ± 7.92	4 M	MD −0.73 95% CI [−0.96, −0.50]	Helpful in lowering HbA1c levels in the intervention group through direct communication with the diabetic patients.
* Souza (2017) Brazil [41]	(A) Community health worker educational program (*n* = 62)	(B) Control group (*n* = 56)	(A) 62.6 ± 11.2 (B) 58.9 ± 11.5	4 M	MD −0.50 95% CI [−1.31, 0.31]	A significant decrease in HbA1c was observed during patients’ follow-up, but it was similar in the intervention and control groups.
Spencer (2011) USA [42]	(A) Community health worker intervention (*n* = 56)	(B) Control group (*n* = 57)	NR	6 M	NR	This study contributes to the growing evidence for the effectiveness of community health workers and their role in multi-disciplinary teams engaged in culturally appropriate health care delivery.
* Stone (2010) USA [43]	(A) Active care management with home telemonitoring (ACM + HT) (*n* = 59)	(B) Monthly care coordination telephone call (CC) (*n* = 69)	NR	3 M/6 M	MD −0.70 95% CI [−1.13, −0.27]	Compared with the CC group, the ACM + HT group demonstrated significantly greater reductions in A1C by 3 and 6 months.
* Takahashi (2004) Japan [44]	(A)Simple new education group (*n* = 15) (C) Long pt. simple education (*n* = 19)	(B) Conventional education group (*n* = 15) (D) Long patient conventional education group (*n* = 19)	New Pt (A) 67.4 ± 8.0 (B) 67.1 ± 8.0 Long Pt (A) 74.4 ± 6.0 (B) 74.2 ± 5.3	3 M/6 M/12 M	New MD −0.10 95% CI [−0.69,0.49] Long MD 0.60 95% CI [−0.02, 1.22]	Simple dietary education is useful and effective for elderly diabetic patients on their first visit in a similar fashion to conventional dietary education. Because of the small effects of both types of education on glucose control in long-term patients, more psychosocial support may be necessary.
* Tamban (2013) Philippine [45]	(A) Short message services (SMS) (*n* = 52)	(B) Control group (*n* = 52)	(A) 48.0 ± 8.1 (B) 51.0 ± 6.2	3 M/6 M	MD −0.35 95% CI [−0.69, −0.01]	The use of SMS as an adjunct to the standard of DM care improved a significant reduction in HbA1c levels after 3 and 6 months.
* Tang (2012) USA [46]	(A) Online with enhanced resources for diabetes (*n* = 186)	(B) Control group (*n* = 193)	(A) 54 ± 10.7 (B) 53 ± 10.2.2	6 M/12 M	MD −0.23 95% CI [−0.58,0.12]	Intervention patients achieved greater decreases in A1C at 6 months than control patients, but the differences were not sustained at 12 months. More intervention group patients than control patients achieved improvement in A1C (>0.5% decrease).
* Thom (2013) USA [47]	(A) Peer Health coaching (*n* = 122)	(B) Control group (*n* = 114)	(A) 56.3 ± 10.3 (B) 54.1 ± 10.4	6 M	MD −0.57 95% CI [−1.12, −0.02]	Peer health coaching significantly improved diabetes control in this group of low-income primary care patients.
Varney (2014) Australia [48]	(A) Telephone coaching (*n* = 47)	(B) Control group (*n* = 47)	(A) 59 (56–62) (B) 64 (61–66)	6 M/12 M	NR	Significant interaction effects were observed between group and time at 6 months, demonstrating improvement in HbA1_C_, fasting glucose, diastolic blood pressure, and physical activity. The intervention’s effect on these parameters was not sustained at 12 months.
* Wild (2016) England [49]	(A) Tele monitoring (*n* = 146)	(B) Control group (*n* = 139)	(A) 60.5 ± 9.8 (B) 61.4 ± 9.	9 M	MD −0.50 95% CI [−0.81, −0.19]	Supported telemonitoring resulted in clinically important improvements in the control of glycemia in patients with type 2 diabetes in family practice.

Mean ± SD: mean standard deviation, Median (IQR, inter-quartile range): Median (IQR), MD: mean difference, CI: confidence interval; HF (High fruit), LF (Low fruit), NR: not reported; Studies included a meta-analysis., *: Studies included a meta-analysis.

## Data Availability

Not applicable.

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
