# Peer review of "The Effects of Dietary Education Interventions on Individuals with Type 2 Diabetes: A Systematic Review and Meta-Analysis"

_ijerph, 2021, doi:10.3390/ijerph18168439_

Round 1
Reviewer 1 Report
Dear authors:
Regarding tha manuscript with the title “Systematic Review of the effects of a dietary education intervention on individuals with type 2 diabetes”, I have a major concern that interferes with manuscript results and discussion. There are plenty of studies within the scope of this systematic review that were not included in this paper. In my opinion, the seacrch strategy needs to be changed. Besides, if authors add studies with exercise and psychosocial parameters, the spectrum is too large (see “lifestyle intervention” OR “behavioral intervention” combined with “diabetes”. Thus, and giving the pertinence of this subject and the gap that exists in literature, I suggest authors to limit the scope of the study only to dietary interventions in comparison with a control group. The analysis of subgrouos regarding the time of intervention and delivery method must continue to be presented.
Below I give some examples of manuscripts that were within eligibility criteria defined by authors and were not included in this paper.
Combination of the word “Nutrition” with “Education”
Li, Y. E., Xu, M., Fan, R., Ma, X., Gu, J., Cai, X., Liu, R., Chen, Q., Ren, J., Mao, R., Bao, L., Zhang, Z., Wang, J., & Li, Y. (2016). The effects of intensive nutrition education on late middle-aged adults with type 2 diabetes.
Combination of the word “Nutritional” with “Program”
Hansel B, Giral P, Gambotti L, Lafourcade A, Peres G, Filipecki C, Kadouch D, Hartemann A, Oppert JM, Bruckert E, Marre M, Bruneel A, Duchene E, Roussel R. A Fully Automated Web-Based Program Improves Lifestyle Habits and HbA1c in Patients With Type 2 Diabetes and Abdominal Obesity: Randomized Trial of Patient E-Coaching Nutritional Support (The ANODE Study). J Med Internet Res. 2017 Nov 8;19(11)
Combination of the word “Nutrition” or “Nutritional” with “Intervention”
Chee, W. S. S., Gilcharan Singh, H. K., Hamdy, O., Mechanick, J. I., Lee, V. K. M., Barua, A., Mohd Ali, S. Z., & Hussein, Z. (2017). Structured lifestyle intervention based on a trans-cultural diabetes-specific nutrition algorithm (tDNA) in individuals with type 2 diabetes: A randomized controlled trial. BMJ Open Diabetes Res Care, 5(1), e000384.
Coppell, K. J., Kataoka, M., Williams, S. M., Chisholm, A. W., Vorgers, S. M., & Mann, J. I. (2010). Nutritional intervention in patients with type 2 diabetes who are hyperglycaemic despite optimised drug treatment–Lifestyle Over and Above Drugs in Diabetes (LOADD) study: Randomised controlled trial. British Medical Journal, 341, c3337.
Yannakoulia, M., Poulia, K. A., Mylona, E., & Kontogianni, M. D. (2007). Effectiveness of an intensive nutritional intervention in patients with type 2 diabetes mellitus: Results from a pilot study. The Review of Diabetic Studies, 4(4), 226–230
Combination the word “Dietary” with “Intervention”
Glasgow, R. E., Nutting, P. A., Toobert, D. J., King, D. K., Strycker, L. A., Jex, M., O'Neill, C., Whitesides, H., & Merenich, J. (2006). Effects of a brief computer-assisted diabetes self-management intervention on dietary, biological and quality-of- life outcomes. Chronic Illness, 2(1), 27–38
Other manuscripts that have results beyond the scope of the systematic review and were not included in this systematic review:
Deakin, T. A., Cade, J. E., Williams, R., & Greenwood, D. C. (2006). Structured patient education: The diabetes X-PERT Programme makes a difference. Diabetic Medicine, 23(9), 944–954
Huang, M. C., Hsu, C. C., Wang, H. S., & Shin, S. J. (2010). Prospective randomized controlled trial to evaluate effectiveness of registered dietitian-led diabetes management on glycemic and diet control in a primary care setting in Taiwan. Diabetes Care, 33(2), 233–239.
Rosal, M. C., Ockene, I. S., Restrepo, A., White, M. J., Borg, A., Olendzki, B., Scavron, J., Candib, L., Welch, G., & Reed, G. (2011). Randomized trial of a literacy-sensitive, culturally tailored diabetes self-management intervention for low-income latinos: Latinos en control. Diabetes Care, 34(4), 838–844.
Minor Comments:
Comment 1:
Authors should change the title to “The Effects of Dietary Education Interventions on Individuals with Type 2 Diabetes: A Systematic review and Meta-Analysis”
Comment 2:
In lines 9-10, authors should change “aimed to confirm the effects of dietary intervention education” by “examine the evidence for the efficacy of dietary education interventions”
Comment 3
In lines 10-11, change “type 2 diabetic patients” by “Patients with type 2 diabetes”
Comment 4:
In line 20, authors should change “blood sugar” by “HbA1c”
Comment 5
Lines 28-30. This sentence “In South Korea, diabetes patients account for 13% of the total adult population, and the prevalence is as high as 1 in 3 elderly people over the age of 65 years” can be withdrawn from the manuscript, as the purpose was to make a systematic review and meta analysis and not to do a study with a sample from South Korea
Comment 6
Lines 30-34. I have the same concern I expressed in the previous comment. Authors must provid readers with wordlwide information and not only from South Korea.
Comment 7:
Line 35: Authors should change “Diabetes is difficult to cure and results in complications” by “Diabetes is a chronic disease and results in complications”. Authors must add a sentence with the major complications of diabetes.
Comment 8:
Before line 44, authors must presente why they focus this stdudy on type 2 diabetes.
Comment 9
Lines 45-46:
Authors should withdrawn this sentence “including one article published in Korea and four articles published in other countries” as it does not bring relevant information to this study.
Comment 10
Line 47-48: “Low carbohydrate diet” is not an intervention method

Author Response
please see the attacment

Reviewer 2 Report
This is a systematic review with meta-analysis of the effect of diabetes education on glycemic control measured with HbA1c.
This is a well-studied subject, so the authors must clarify very well what they contribute again to the extensive bibliography that exists.
The inclusion criteria are not clear to me. I suppose you accept studies that compare diabetes education versus no education? Standard treatment? I do not see it clearly. It should better explain the characteristics of the control group.
In the section "Data extraction and quality assessment" I think there must be an error. They do not explain what I expected but refer to clinical situations that I suppose are the exclusion criteria.
In short, there should be a clear section of inclusion and exclusion criteria.
I assume that Figure 3 refers to the effect on HbA1c (not indicated in the title of Figure 3).
Figures 3 and 4 are redundant. Figure 4 includes the information from figure 3 but separating the articles by the duration of diabetes education. I think figure 3 is superfluous.
I'm sorry but the difference between figures 4 and 5 is not clear to me.
They should be more explicit titles.
Reviewer 3 Report
The paper is a systematic review of randomized controlled trials aimed to analyze the effect of different dietary education interventions on the control of diabetes. Thirty-three studies out of the 36 included in the analysis, were used in a meta-analysis to estimate the effect size between the dietary education and general intervention in patients with type 2 diabetes (T2DM), as well as in subgroups considering the application period, intervention methods and intervention contents.
The authors discuss an important number of limitations about the diversity of contents and methods of interventions among studies, the exclusion of some research reports, and the possibility of publication bias or overestimation of the results. In any case, they conclude that regarding to the education method, an intervention combining diet, exercise and psychosocial items in an individualized manner for at least 3 months is highly effective in controlling blood sugar levels.
The impact of this article may be important, due to the high prevalence and mortality of T2DM worldwide, and the importance that basic intervention through dietary education may have for the control of diabetes. In general, the quality and depth of the study is promising, however, a revision of the following aspects would be advisable: authors should revise the English style and language (for example, lines 133, 138…) and clarify some writing concepts (for example, authors should explain the meaning of * in Table 1).
Round 2
Reviewer 1 Report
Dear authors:
Regarding the manuscript with the title “Systematic Review of the effects of a dietary education intervention on individuals with type 2 diabetes”, you give convincing responses to my major concerns.
I only have one minor comment to make regarding the paper in its actual form.
In line 134, regarding exclusion criteria, authors stated “types of intervention other than dietary education”. One of the conclusions of this systematic review is that combining diet, exercise and psychosocial intervention is more effective than diet education alone. Thus, authors must check the intention they have with this sentence and rephrase it.

Author Response
According to the reviewer's comments, the contents have been clearly rephrased as follows. Changes are highlighted in green.
The exclusion criterion was 'educational intervention that did not include dietary intervention education' as an intervention, and all studies that provided dietary intervention education and other education at the same time were included. To express this, the exclusion criteria were clarified, the types of interventions in the selected research literature were explained in the research results, and subgroup analysis was performed.
lines 134-135.
“(3) educational interventions that do not include dietary interventions,”
lines 153-157.
“Of the 33 literatures available for meta-analysis, 5 studies provided only dietary education interventions, the other 9 studies provided dietary education interventions and exercise therapy, and the remaining 19 studies provided dietary education interventions, exercise and psychosocial therapy.”
lines 322-326.
“There were seven studies on dietary-centered education interventions. Interventions in the group included a low-carbonate group, low-fat group, low glycemic index (GI) diet group, and low-fruit group vs. high-fruit group. There were nine dietary and exercise education interventions and 19 studies on dietary, exercise, and psychosocial education interventions (Fig. 6)”
